# Practice Guidelines of the Central European Hepatologic Collaboration (CEHC) on the Use of Thrombopoietin Receptor Agonists in Patients with Chronic Liver Disease Undergoing Invasive Procedures

**DOI:** 10.3390/jcm10225419

**Published:** 2021-11-19

**Authors:** Robert Flisiak, Krasimir Antonov, Pavel Drastich, Peter Jarcuska, Marina Maevskaya, Mihály Makara, Željko Puljiz, Borut Štabuc, Anca Trifan

**Affiliations:** 1Department of Infectious Diseases and Hepatology, Medical University of Białystok, Zurawia 14, 15-540 Białystok, Poland; 2Department of Gastroenterology, University Hospital ‘St. Ivan Rilski’, 1431 Sofia, Bulgaria; krasi_antonov@abv.bg; 3Department of Hepatogastroenterology, Institute for Clinical and Experimental Medicine, 140 21 Prague, Czech Republic; pavel.drastich@ikem.cz; 4Second Department of Internal Medicine, Faculty of Medicine and L. Pasteur University Hospital, P.J. Safarik University, Trieda SNP 1, 040 11 Kosice, Slovakia; peter.jarcuska@upjs.sk; 5Clinic of Propedeutics of Internal Diseases, Gastroenterology and Hepatology Named after V. Kh. Vasilenko, Federal State Autonomous Educational Institution of Higher Education I.M. Sechenov First Moscow State Medical University (Sechenov University) of the Ministry of Health of the Russian Federation, 119435 Moscow, Russia; liver.orc@mail.ru; 6Central Hospital of Southern Pest National Institute of Haematology and Infectious Diseases, 1097 Budapest, Hungary; michael@makara.md; 7Department of Gastroenterology and Hepatology University Hospital Split, Split School of Medicine, 21000 Split, Croatia; zpuljiz4@gmail.com; 8Division of Internal Medicine, University Medical Centre Ljubljana, 1000 Ljubljana, Slovenia; borut.stabuc@kclj.si; 9Department of Internal Diseases, Institute of Gastroenterology and Hepatology Lasi, University of Medicine and Pharmacy “Gr. T. Popa”, 700115 Lasi, Romania; ancatrifan@yahoo.com

**Keywords:** chronic liver disease, avatrombopag, thrombocytopenia, surgical procedures, thrombopoietin receptor agonists, platelet transfusion

## Abstract

Background: Second-generation thrombopoietin receptor agonists (TPO-RAs) are emerging as the new standard for managing thrombocytopenia (TCP) in patients with chronic liver diseases (CLDs) undergoing scheduled procedures. However, practical guidance for their routine use in CLD patients undergoing specific invasive procedures is lacking. Methods: These practice guidelines were developed by the Initiative Group for Central European Hepatologic Collaboration (CEHC), composed of nine hepatologist/gastroenterologist experts from Central Europe. Using an adapted Delphi process, the CEHC group selected ten invasive procedures most relevant to the hepatology/gastroenterology setting in the region. Consensus recommendations for each invasive procedure are reported as a final percentage of expert panel responses. Results: A consensus was agreed that TPO-RAs should be considered for raising platelet count in CLD patients undergoing scheduled abdominal surgery, high-bleeding risk dentistry, endoscopic polypectomy, endoscopic variceal ligation, liver biopsy, liver surgery, liver transplantation and percutaneous ablation, but it was also agreed that they are less beneficial or not necessary for endoscopy without intervention and paracentesis. Conclusions: Using a modified Delphi method, experts reached an agreement for TCP management in CLD patients undergoing ten invasive procedures. These practice guidelines may help with decision making and patient management in areas where clinical evidence is absent or limited.

## 1. Introduction

Chronic liver diseases (CLDs) are a substantial and underestimated public health burden associated with high mortality [1]. Worldwide, 844 million people have a CLD, which has a mortality rate of two million deaths per year [1]. Progression of CLD to fibrosis and end-stage cirrhosis, liver failure and hepatocellular carcinoma is associated with increased patient morbidity, hospitalization frequency and deteriorating quality of life [2,3]. Reduced hepatic production of thrombopoietin (TPO) together with direct bone marrow suppression are key factors in the development of thrombocytopenia (TCP) in liver cirrhosis, resulting in decreased megakaryocyte stimulation and platelet production [4]. Furthermore, TCP level is a predictive parameter of bleeding risk in CLD, especially regarding risk of hemorrhagic events in cirrhotic patients [5]. 

TCP, defined as a platelet count less than 150 × 10^9^/L, is the most common hematological complication associated with CLDs, affecting up to 76% of patients with advanced fibrosis or liver cirrhosis [4,6,7,8,9,10]. Compared to non-cirrhotic patients, those with cirrhosis are almost 12 times more likely to have at least moderate TCP, i.e., a platelet count less than 100 × 10^9^/L [11]. An analysis by the Acute Liver Failure Study Group enrolling 1600 patients documented that the median platelet count for liver disease patients following hospital admission was approximately 130 × 10^9^/L [5]. Notably, 60% of patients analyzed in the study had mild TCP (platelet count < 150 × 10^9^/L), 35% had moderate TCP (platelet count < 100 × 10^9^/L) and 10% had severe TCP (platelet count < 50 × 10^9^/L) [5].

Platelet transfusion is commonly used for the clinical management of TCP in patients with CLD undergoing invasive procedures. However, the use of platelet transfusion may be limited by the development of antiplatelet antibodies, high costs, short duration of storage and efficacy, risk of infection, and other transfusion-related risks and complications [12]. Platelet transfusions also rely on donors and are given intravenously [13]; therefore, they are often avoided due to a lack of clear beneficial effect and their potential for side effects [14]. Oral TPO receptor agonists (TPO-RAs) are an alternative management option that can be used before surgery to stimulate TPO and increase platelet count, thus avoiding the requirement for platelet transfusions [12,13]. Moreover, TPO-RAs mitigate preprocedural thrombocytopenia in patients with CLD by raising platelet count for longer periods (+3 weeks). They are also more predictable in increasing platelet count and are taken at home, so their use reduces resource wastage and hospital stays [13,14].

Two oral TPO receptor agonists, avatrombopag [15] and lusutrombopag [16], are available in Europe to treat severe thrombocytopenia in adult patients with CLD scheduled to undergo a surgical procedure. Avatrombopag and lusutrombopag are taken more than one week before an invasive intervention so that they can be used only for planned procedures [13]. Due to the risk of transfusion-related complications, TPO-RAs offer a cost-effective treatment choice for many treating physicians in Central European healthcare systems relevant to the rest of Europe [13]. However, clinical practice guidance based on procedure risks and appropriate platelet targets using TPO-RA agents is lacking. 

The goal of these practice guidelines of the Initiative Group for Central European Hepatologic Collaboration (CEHC) is to provide expert opinions and evidence-based, risk-adapted recommendations to help physicians better manage thrombocytopenia using TPO-RAs in CLD patients undergoing elective surgical interventions, reduce the need for platelet transfusions, and decrease the risk of bleeding in CLD patients with concurrent TCP before scheduled procedures. 

## 2. Summary of Clinical Evidence for Thrombopoietin Receptor Agonists (TPO-RAs)

Table 1 summarizes the key phase 3 clinical trials of avatrombopag and lusutrombopag for the discussed indication [15,16,17]. The practice-changing safety and efficacy data from the phase 3 trials will help inform decision making and management of TCP in patients with CLD undergoing a scheduled procedure. Based on these pivotal data, the European Medicines Agency (EMA; https://www.ema.europa.eu/en, accessed on 1 September 2021) authorized avatrombopag to treat severe TCP in adult patients with CLD scheduled to undergo an invasive procedure in June 2019 and for lusutrombopag in February 2019.

Avatrombopag is an oral, small-molecule TPO-RA developed to provide a predictable increase in platelets as an alternative to platelet transfusions [15]. Two identical, multicenter, randomized placebo-controlled phase 3 trials (ADAPT-1 and ADAPT-2) were conducted to evaluate the safety and efficacy of avatrombopag in CLD patients with TCP [18,19]. The study design and patient populations of the two studies have been previously described [18,19]. ADAPT-1 and ADAPT-2 enrolled 435 patients and represent the largest published data set for TPO-RAs in the CLD patient population [19]. Pooled phase 3 data from ADAPT-1 and ADAPT-2 showed that avatrombopag was superior to placebo overall and in the baseline platelet count subgroups, since a higher proportion of avatrombopag-treated patients in ADAPT-1 and ADAPT-2 did not require a platelet transfusion or rescue procedure for bleeding (Table 1 and Figure 1) [18,19]. The treatment differences were both clinically meaningful and statistically significant (*p* < 0.0001) [18,19]. Platelet count increase was observed from day 4 in ADAPT-1 and ADAPT-2, regardless of baseline platelet count, reaching a maximum level at days 10-13 [18,20]. The mean platelet count remained at or above 50 × 10^9^/L at day 17, with three patients reaching a platelet count more than 200 × 10^9^/L [18]. Safety analyses have also been previously reported, demonstrating that avatrombopag was well tolerated and comparable to the placebo arm [18,19].

Lusutrombopag is another oral, small-molecule TPO agonist that stimulates platelet production through its action on TPO surface cells of megakaryocytes [16]. Evidence supporting the efficacy and safety of lusutrombopag is provided from two multicenter, randomized, double-blind, parallel-group, placebo-controlled phase 3 studies, L-PLUS 1 [21] and L-PLUS 2 [22]. The primary outcomes for L-PLUS 1 and L-PLUS 2 were similar to the phase 3 trials for avatrombopag (Table 1 and Figure 1). Pooled data from L-PLUS 1 and L-PLUS 2 showed that lusutrombopag vs. placebo was associated with a numerically lower rate of postprocedural bleeding (6.7% vs 10.6%, respectively) without increased risk of thrombosis [21,22,23]. In addition, adverse events were somewhat balanced between the treatment and placebo arms [21,22,23].

Based on the strength of evidence from clinical studies, recent guidelines from the British Society of Gastroenterology [24], and treatment algorithms from experts in the U.S. [17] and Canada [25] recommend using TPO-RAs as an alternative to platelet transfusion according to local protocol. Notably, only a few studies, among those that assessed the risk of bleeding in relation to platelet count, found that TCP may be predictive of bleeding following percutaneous liver biopsy, dental extractions, percutaneous ablation of liver tumors and endoscopic polypectomy [20].

## 3. Methods

A modified Delphi process was adopted to develop consensus guidelines according to the clinical importance of invasive procedures within Central Europe. In an area lacking certainty, the Delphi method uses multiple rounds of structured feedback to achieve consensus [26]. Following a virtual advisory board meeting on 22 February 2021, a questionnaire describing procedure-related platelet count thresholds in patients with cirrhosis and severe thrombocytopenia was developed. The questionnaire was discussed and refined during a virtual follow-up meeting on 2 June 2021 before being circulated by email to nine representative CEHC group members. 

The questionnaire focusses on ten routine invasive procedures grouped into three main types of intervention: (1) endoscopic/endovascular procedures (endoscopic polypectomy, endoscopic variceal ligation, endoscopy without intervention (e.g., gastroscopy, colonoscopy) and percutaneous ablation); (2) surgical procedures (abdominal surgery and other invasive procedures (e.g., vascular catheter insertion, HVPG measurement, cholecystectomy, herniotomy, thoracentesis, urological surgery, other), paracentesis, liver biopsy, liver surgery and liver transplantation); and (3) dentistry (high-bleeding-risk dentistry (e.g., tooth extraction, root canal procedures, dental implants and comprehensive hygienist procedures). Anonymized questionnaire responses were collected and analyzed by two independent reviewers, then emailed back to all nine CHEC guideline development group members for second-round review. Due to an absence of regional and international consensus statements and guidelines on TPO-RA use for CLD patients with TCP undergoing elective procedures, the expert CEHC group used the European systematic literature review recently conducted by Alvaro et al. (2021) [20], where appropriate, as evidence to support each platelet count threshold recommendation. Good practice recommendations were also formulated based on the clinical experience of the CHEC guideline development group. A consensus was considered to have been reached when all nine CEHC group members had no further substantive comments and approved the threshold recommendations for publication. Agreed platelet count thresholds for each invasive procedure are reported as a final percentage based on the questionnaire responses of the experts.

## 4. Results

Consensus results for target platelet count and use of TPO-RAs in CLD patients with TCP undergoing specific procedures are shown in Table 2. Overall, the CEHC experts reached a consensus that five procedures (abdominal surgery, endoscopic polypectomy, liver biopsy, liver surgery, and percutaneous ablation) are not recommended for CLD patients with a platelet count < 50 × 10^9^/L (<80 × 10^9^/L for liver surgery). Most experts (88.9%) agreed that high-risk dentistry might be performed for platelet count > 50 × 10^9^/L. In addition, the experts agreed that TPO-RAs are beneficial for raising platelet count in CLD patients before abdominal surgery (100.0%), high-bleeding risk dentistry (100.0%), endoscopic polypectomy (88.9%), endoscopic variceal ligation (88.9%), elective liver biopsy (100%), liver surgery (100.0%), liver transplantation (77.8%) and percutaneous ablation (100.0%), with only approximately half of the experts considering this a therapeutic modality for endoscopy without intervention (44.4%) and paracentesis (55.6%). A treatment algorithm for CLD patients with TCP scheduled to undergo an invasive procedure was developed (Figure 2).

### 4.1. General Considerations and Comments

Management of TCP for CLD patients requiring invasive procedures should be categorized for either long-term or short-term treatment. TPO-RA is likely to be the best short-term management solution and should be considered for all CLD patients with a platelet count ≤ 50 × 10^9^/L. Patients with a Model of End-stage Liver Disease (MELD) score greater than 20 should be first referred for liver transplantation. TPO-RA treatment is recommended for acute alcoholic hepatitis patients requiring an urgent, high-risk bleeding surgical procedure to protect them from thrombocytopenia. Even if the patients become abstinent, the platelet count can rise slowly over a minimum of several months or even a few years. However, rapid improvement or normalization of platelet count can be achieved for some patients with acute alcoholic hepatitis.

Following the initial advisory board and follow-up discussions, it was noted that endoscopy, paracentesis, and thoracentesis are the most common procedures performed by healthcare providers (HCPs) in CLD patients with TCP. For such procedures, the major drawback of using platelet transfusions is the frequent issue in Central European countries of a lack of blood products, which has been further impacted due to the recent COVID-19 pandemic [27]. In contrast, TPO-RAs are orally administered, relatively easy to store and dispense, and are typically less susceptible to supply chain issues [28].

### 4.2. Contraindications for the Use of TPO-RAs

Thrombopoietin analogues should not be used or should be used with great caution in patients with a history of thrombotic events [14,15,16]. These treatments are also not recommended in patients who have portal vein thrombosis [14,15,16]. Due to the increased thrombotic potential and lack of robust clinical data, the CEHC experts do not currently recommend TPO-RAs for patients with coronavirus disease 2019 (COVID-19) [29].

## 5. Discussion

Since randomized, placebo-controlled clinical trials are not available on the use of TPO-RAs in the different surgical procedures relating to CLD, other methods of obtaining reliable information are required. Therefore, a modified Delphi technique was selected to obtain consensus practice recommendations of CEHC experts from Central Europe. 

There are limited data to inform bleeding risk following invasive procedures in patients with advanced liver disease and thrombocytopenia [20]. A recent review by the Italian Procedure-Related bleeding Risk in Cirrhosis (PReBRIC) group showed substantial variability in the use of prophylactic platelet transfusions across the country [20]. The PreBRIC group also reported that definitive conclusions based on evidence from the literature about appropriate target platelet count to improve the risk of bleeding in cirrhotic patients who underwent invasive procedures are not possible [20]. While the CEHC experts from across Central Europe acknowledge that the platelet count at which a given procedure carries an acceptable risk of bleeding is unique for each patient and procedure, in this paper, we provide practical guidance based on the latest evidence from the literature and personal clinical experience.

CLD patients require multiple routine invasive procedures such as transjugular liver biopsy, transarterial chemoembolization and transarterial radioembolization over the course of their disease, which utilizes significant medical resources and incurs high medical care costs [30,31,32]. Notably, patients with CLD and moderate/severe TCP are at increased risk of bleeding when undergoing elective or urgent invasive procedures [13]. Furthermore, TCP is related to poorer outcomes for CLD patients, such as decreased quality of life, morbidity from untreated acute problems, postponed therapy for chronic conditions and increased risk of death while awaiting transplant [8,31]. Currently, spontaneous and clinically significant bleeding is relatively uncommon in patients with acute liver failure, which may be attributed to hemostatic and intensive care management improvements over the past decades [33]. This observation is confirmed in a recent analysis by Stravitz et al. 2018 (Dutch Study Group) of 1770 adult patients with acute liver failure [34]. This study reported bleeding complications in only 11% of patients following hospital admission; bleeding complications were the proximate cause of death in only 5% of cases [34]. However, TCP still impacts routine care since many CLD patients with TCP may be ineligible for surgical procedures due to increased risk of bleeding [8]. While some studies have found no increase in the risk of bleeding in patients with platelet counts more than 50 × 10^9^/L undergoing these procedures [20,35,36], many physicians postpone or avoid invasive procedures (e.g., dental procedures) associated with a risk of bleeding in patients with CLD and TCP [36]. New second-generation TPO-RAs recently approved by the EMA, lusutrombopag and avatrombopag, provide an alternative solution to platelet transfusions for TCP management associated with CLD [36].

TPO-RAs may potentially reduce the risk of invasive interventions and allow access to scheduled procedures with a reduced hospital stay, reduced risk for transfusion-associated complications and improved quality of life for CLD patients. Thus, there is an urgent need to develop practice-related recommendations for using TPO-RAs for this patient population, considering clinical and health economic implications. A recent TPO-RA health technology assessment by NICE in the U.K. concluded that avatrombopag and lusutrombopag have positive implications in clinical practice [13]. However, avatrombopag is also indicated to treat chronic immune thrombocytopenia (CIT), so its clinical use is expected to be broader and involve consultant hematologists [15].

Our work has several strengths. First, the CEHC Initiative Group included esteemed experts in hepatology/gastroenterology across eight different Central European countries, with many years of experience managing CLD patients with TCP. Second, the anonymity of the experts’ responses was preserved until completion to avoid inherent bias during the modified Delphi process due to dominance and group pressure. Third, we completed the process over two survey rounds and achieved agreement (6/9 experts or more than 66.7% consensus) on using TPO-RAs in 80% of the ten invasive procedures selected. Our work also has several limitations. The consensus guidelines were compiled by a small group of hepatologists/gastroenterologists and may not represent the views of all experts in the field or for all Central European countries. The CEHC guidelines were developed based on available evidence and personal experience; however, clinical trial and real-world data would help strengthen our recommendations. Ten common procedures were evaluated in our study, but many other invasive procedures were not included. The study did not take regional differences in healthcare infrastructure and resources into account. The feedback from the experts was anonymized in all rounds to minimize the risk of responder bias; however, it is possible that the responses could have been influenced by the way the individual experts interpreted the different invasive procedures. Moreover, different levels of experience with TPO-RAs, procedures and variations in national practices may also have influenced experts’ opinions.

## 6. CEHC Recommendations for Using TPO-RA Therapy Prior to Scheduled Invasive Procedures

TPO-RA use and platelet count threshold recommendations for CLD patients with TCP scheduled to undergo an invasive procedure are shown in Table 2 and depicted in Figure 2 as a treatment algorithm.

### 6.1. Recommendations for Use of TPO-RAs

**Statement** **1.**We recommend that TPO-RAs be considered for all patients undergoing elective endoscopic/endovascular, surgical, and high-bleeding-risk dentistry procedures, as described in Table 2.

Treatment with TPO-Ras may or may not be necessary for patients undergoing planned endoscopy without intervention (e.g., gastroscopy, colonoscopy) or paracentesis; thus, a personalized therapy approach is recommended. TPO-RAs should always be considered for patients with Child Pugh score C.

### 6.2. Recommendations for Platelet Count Threshold

**Statement** **2.**We recommend planned endoscopic variceal ligation, endoscopy without intervention, liver transplantation and paracentesis for patients with severe or moderate TCP, i.e., platelet count more than 30 × 10^9^/L. Paracentesis may be safe even in patients with platelet counts below 30 × 10^9^/L.

**Statement** **3.**We recommend planned liver biopsy for patients with moderate TCP only, i.e., platelet count more than 50 × 10^9^/L, except for patients with portal hypertension when platelet count should be more than 80 × 10^9^/L.

**Statement** **4.**We generally do not recommend elective liver surgery in patients with a platelet count below 80 × 10^9^/L.

## 7. Conclusions

These practice recommendations and the treatment algorithm will help guide hepatologists/gastroenterologists routinely managing TCP in CLD patients. By potentially reducing transfusion-associated complications and improving patient quality of life for patients, avatrombopag or lusutrombopag may become the treatment of choice in elective surgical interventions in Central European countries and elsewhere. However, several questions regarding the use of TPO-RAs and platelet transfusions for both elective and urgent procedures remain unanswered. Therefore, obtaining new evidence from clinical trials and real-world settings should be prioritized to help better position thrombopoietin analogs as the new standard treatment for CLD patients with TCP undergoing invasive procedures.

## Figures and Tables

**Figure 1 jcm-10-05419-f001:**
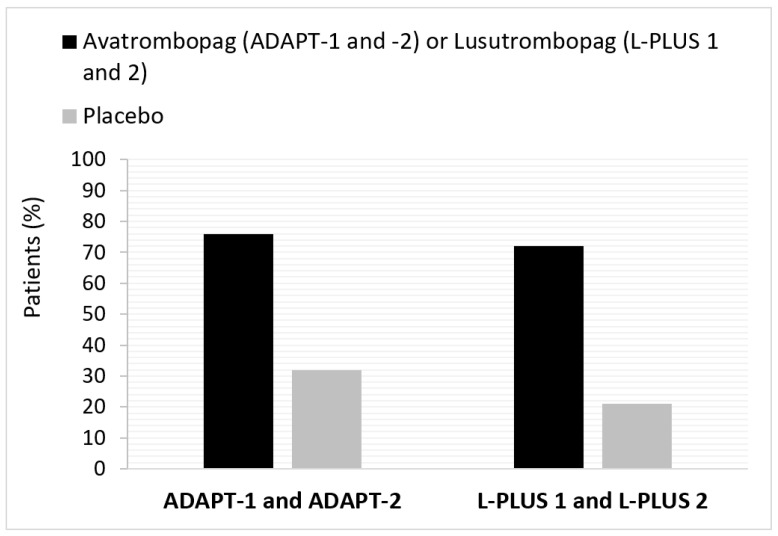
Pooled responders not requiring a platelet transfusion prior to an invasive procedure in ADAPT-1 and ADAPT-2 (avatrombopag) and L-PLUS 1 and L-PLUS 2 (lusutrombopag). Responders are defined as the subjects who achieved platelet count ≥ 50 × 10^9^/L on the day of the procedure. ADAPT-1/ADAPT-2 [18,19] and L-PLUS 1 [21]/L-PLUS 2 [22] are phase 3 trials for avatrombopag and lusutrombopag, respectively.

**Figure 2 jcm-10-05419-f002:**
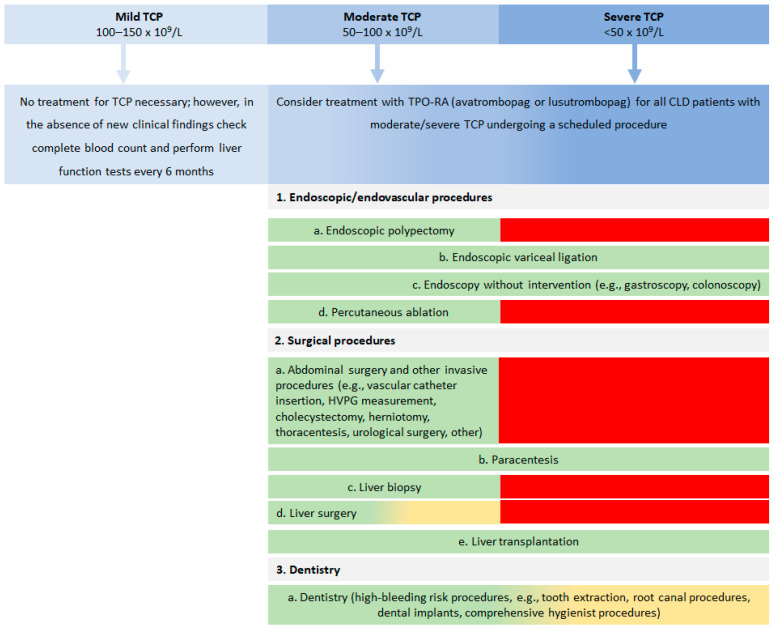
Consensus treatment algorithm for the use of TPO-RAs and platelet count threshold in CLD patients with TCP scheduled to undergo an invasive procedure. Contraindications for TPO-RA treatment include patients with a history of thrombotic events, portal vein thrombosis, and COVID-19 infection. Color code: green = procedure can be performed, yellow = procedure may be considered and red = procedure is generally not recommended. CLD, chronic liver disease; COVID-19, coronavirus disease 2019; TCP, thrombocytopenia.

**Table 1 jcm-10-05419-t001:** Summary of Phase 3 Clinical Trial Efficacy Results of Avatrombopag and Lusutrombopag for the Treatment of TCP in Patients with CLD Undergoing Invasive Procedures.

Study (Publication Year)	Participants	Gender	Age (Years)	Interventions	Mean Baseline Platelet Count ×10^9^/L (Mean ± S.D.)	Primary Efficacy Outcome Measure	Summary of Key Efficacy Results	Ref.
Avatrombopag—Phase 3 trials:								
ADAPT-1 (2018)	*N* = 231	M: 68.4% F: 31.6%	56.35 ± 9.52 56.22 ± 1.05	avatrombopag vs. placebo treatment for 5 days	36.15 ± 8.58 36.80 ± 8.96	% patients who did not require a platelet transfusion or rescue procedure for bleeding following randomization and up to 7 days after a scheduled procedure	Overall (*N* = 435): Responders 75.8% avatrombobag vs. 31.7% placebo (treatment difference * Δ44.2; 95% CI: 35.3, 53.0; *p* < 0.0001)	Terrault et al., 2018; Poordad et al., 2020 [18,19]
							Low baseline platelet count subgroup (<40 × 10^9^/L; *n* = 251): Responders 66.9% avatrombobag vs. 28.6% placebo (treatment difference * Δ38.3; 95% CI: 26.5, 50.1; *p* < 0.0001)	
ADAPT-2 (2020)	*N* = 204	M: 62.3% F: 37.7%	58.28 ± 2.84 58.13 ± 1.25	avatrombopag vs. placebo treatment for 5 days	37.98 ± 7.14 38.21 ± 7.74		High baseline platelet count subgroup (≥40 to <50 × 10^9^/L; *n* = 184): responders 88.0% avatrombobag vs. 35.8% placebo (treatment difference * Δ52.2; 95% CI: 39.3, 65.1)	
Lusutrombopag—Phase 3 trials:								
L-PLUS 1 (JapicCTI-132323; 2019)	*N* = 97 (49 lusutrombopag; 48 placebo)	M: 53.1% F: 46.9%	67.8 ± 8.60	lusutrombopag vs. placebo treatment for up to 7 days	40.4 ± 6.60 (17.7% <35 × 10^9^/L; 53.1% ≥35 to <45 × 10^9^/L; 29.2% >45 × 10^9^/L)	% patients who did not require a platelet transfusion before the primary invasive procedure	Overall (*N* = 97): Responders 79.2% lusutrombopag vs. 12.5% placebo (treatment difference Δ66.7; *p* < 0.0001)	Hidaka et al., 2019 [20]
L-PLUS 2 (2019)	*N* = 215 (108 lusutrombopag; 107 placebo)	M: 62.3% F: 37.7%	51.8 ± 11.3	lusutrombopag vs. placebo treatment for up to 7 days	37.55 (34.4% <35 × 10^9^/L; 64.7% ≥35 × 10^9^/L)	% patients who did not require a platelet transfusion or rescue procedure for bleeding following randomization and up to 7 days after a scheduled procedure	Overall (*N* = 215): Responders 64.8% lusutrombopag vs. 29.0% placebo (treatment difference Δ36.7%; 95% CI: 24.9, 48.5; *p* < 0.0001)	Peck-Radosavljevic et al., 2019 [21]
							Low baseline platelet count subgroup (<35 × 10^9^/L; *n* = 74): responders 41.7% lusutrombopag vs. 18.4% placebo (treatment difference Δ23.3)	
							High baseline platelet count subgroup (≥35 × 10^9^/L; *n* = 139): responders 77.5% lusutrombopag vs. 33.8% placebo (treatment difference Δ43.7)	

Responders are defined as the subjects who achieved platelet count ≥50 × 10^9^/L on the procedure day. *p*-value is based on the Wilcoxon rank-sum test for each avatrombopag treatment group versus placebo within each baseline platelet count subgroup. * Treatment difference = proportion of responders for avatrombopag−proportion of responders for placebo; 95% confidence interval is calculated based on normal approximation. Abbreviations: CLD, chronic liver disease; M, male; F, female; N.R., not reported; *p*, probability value; TCP, thrombocytopenia.

**Table 2 jcm-10-05419-t002:** CEHC Initiative Group consensus recommendations for managing TCP in patients with CLD scheduled for an invasive procedure.

Procedure	Benchmark *	Minimum Platelet Count for Procedure *n* (%)			Is TPO-RA Suitable for Platelet Count Elevation? *n* (%)	Additional Comments and Considerations
		>30 × 10^9^/L	>50 × 10^9^/L	>80 × 10^9^/L		
1. Endoscopic/endovascular procedures:						
a. Endoscopic polypectomy	Bleeding risk ~7.5% for patients with platelet count < 50 × 10^9^/L (retrospective data); Immediate post-procedural bleeding rate was 27.5% with RR = 6	NR	9 (100.0%)		Yes: 8 (88.9%) No: 1 (11.1%)	
b. Endoscopic variceal ligation	Bleeding risk ~2.75−7.33%; No association between bleeding risk and platelet count	7 (77.8%)	2 (22.2%)		Yes: 8 (88.9%) No/NA: 1 (11.1%)	TPO-RA can be used for urgent procedures regardless of platelet count; For elective ligation, TPO-RA is recommended when platelet count is <50 × 10^9^/L; In acute variceal bleeding, ligation may be performed at any platelet count, i.e., as secondary prophylaxis when platelet count is >30 × 10^9^/L
c. Endoscopy without intervention (e.g., gastroscopy, colonoscopy)	No data was provided in the article; Advisory Board discussed the low risk of bleeding	9 (100%)			Yes: 4 (44.4%) No/NA: 5 (55.6%)	Not performed in patients with spontaneous bleeding; May be performed at any platelet count
d. Percutaneous ablation	Rarely performed in patients with platelet count < 50 × 10^9^/L and is usually preceded by platelet transfusions and close monitoring of platelet count; Bleeding risk following radio-frequency ablation of HCC is <1	NR	9 (100.0%)		Yes: 97 (100.0%) No: 0 (0.0%)	
2. Surgical procedures:						
a. Abdominal surgery and other invasive procedures **	Available evidence insufficient to assess association between platelet count and post-procedural bleeding risk	NR	8 (88.9%)	1 (11.1%)	Yes: 9 (100.0%) No: 0 (0.0%)	
b. Paracentesis	Typically performed in cirrhotic patients with significant portal hypertension and TCP; No bleeding was recorded in patients with platelet count < 50 × 10^9^/L	9 (100.0%)			Yes: 5 (55.6%) No/NA: 4 (44.4%)	In patients with severe dyspnoea due to large ascites, evacuatory paracentesis is recommended even at lower platelet counts; Paracentesis may be performed at any platelet count; can be safe even if platelet count is <30 × 10^9^/L but can be associated with bleeding in rare situations
c. Liver biopsy	Bleeding risk ~0.6%; Usually performed in patients without portal hypertension and platelet count > 50 × 10^9^/L	NR	8 (88.9%)	1 (11.1%)	Yes: 9 (100%) No: 0 (0%)	For percutaneous liver biopsy; Except for patients with portal hypertension when platelet count should be >80 × 10^9^/L; In the last few years, liver biopsy has become less popular and Central European physicians are more cautious
d. Liver surgery	Portal hypertension is the main determinant of outcome; Even mild TCP (platelet count < 150 × 10^9^/L) predicted major postoperative complications and mortality after resection of HCC	NR	1 (11.1%)	8 (88.9%)	Yes: 9 (100.0%) No: 0 (0.0%)	
e. Liver transplantation	No association between platelet count and intra- or post-transplantation bleeding	7 (77.8%)	1 (11.1%)	1 (11.1%)	Yes: 7 (77.8%) No: 2 (22.2%)	May be performed at any platelet count; Usually not a planned procedure
3. Dentistry:						
a. Dentistry (high-bleeding risk procedures) **	Bleeding risk seemed to be inherently related to the procedure or the number of teeth extracted rather than to platelet count; Bleeding risk ~2.9% for a patient with platelet count = 50 × 10^9^/L and INR =2.5 (prospective study data)	1 (11.1%)	8 (88.9%)		Yes: 9 (100.0%) No: 0 (0.0%)	Local therapy is generally preferred; Patient and procedure dependent; There is currently no uniformity between dentists; Many Central European dentists request platelet transfusions for platelet count < 80 × 10^9^/L; TPO-RAs should always be considered for patients with Child Pugh score C

* Existing Evidence-Based Recommendations from Alvaro et al., 2021 [20] and the Central European Advisory Board on 22 February 2021. ** Abdominal surgery, e.g., vascular catheter insertion, HVPG measurement, cholecystectomy, herniotomy, thoracentesis, urological surgery, other; Dentistry high-risk bleeding procedures, e.g., tooth extraction, root canal procedures, dental implants, comprehensive hygienist procedures. Consensus recommendations reported as a percentage of the total expert responses. Note: Only a few studies that assessed the risk of bleeding in relation to platelet count found that TCP may be predictive of bleeding following percutaneous liver biopsy, dental extractions, percutaneous ablation of liver tumors and endoscopic polypectomy. Procedures are grouped by category of procedure for easy reference rather than in order of the frequency they are performed. Abbreviations: CLD, chronic liver disease; HCC, hepatocellular carcinoma; INR, international normalized ratio; NR, not recommended; RR, relative risk; TCP, thrombocytopenia.

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
