# Peer review of "Practice Guidelines of the Central European Hepatologic Collaboration (CEHC) on the Use of Thrombopoietin Receptor Agonists in Patients with Chronic Liver Disease Undergoing Invasive Procedures"

_jcm, 2021, doi:10.3390/jcm10225419_

Round 1

Reviewer 1 Report

Manuscript is clear, relevant for the field and presented in a well-structured manner. References are current.  Conclusions are consistent with the evidence and arguments presented.

Author Response

We thank Reviewer 1 for their comments. No changes were requested.

Reviewer 2 Report

The authors present their practice guidelines for the use of second-generation thrombopoietin receptor agonists (TPO-RAs) for managing thrombocytopenia (TCP) in patients with chronic liver diseases (CLDs) undergoing invasive procedures.

The practice guidelines were developed by the Initiative Group for Central European Hepatologic Collaboration (CEHC), composed of 9 hepatologist/gastroenterologist experts from Central Europe, using an adapted Delphi process.

The CEHC group selected ten invasive procedures most relevant to the hepatology/gastroenterology setting in the region, and the consensus recommendations for each invasive procedure were finally reported as the percentage of expert panel responses.

The authors conclude that their practice guidelines may help decision-making and patient management in this field, where clinical evidence is absent or limited.

The manuscript is well thought out and well written. The overview of thrombocytopenia in CLD and the summary of clinical evidence for TPO-RAs are exhaustive. The CEHC group finally suggests the clinical behavior and the rationale for the use of TPO-RAs for a number of procedures commonly required in CLD patients, including scheduled abdominal surgery, high-bleeding risk dentistry, endoscopic polypectomy, endoscopic variceal ligation, liver biopsy, liver surgery, liver transplantation, percutaneous ablation, endoscopy without intervention and paracentesis. The discussion is exhaustive.

The main limitation of the study is that the consensus guidelines were compiled by a very small group of hepatologists/gastroenterologists, so that further evidence should be expected from clinical trials and real-world settings.  Thus, authors should better emphasize that their findings are preliminary.

Author Response

We thank Reviewer 2 for their comments. We have added an additional sentence in the discussion section to clarify that further data would help strengthen our recommendations: "The CEHC guidelines were developed based on available evidence and personal experience; however, clinical trial and real-world data would help strengthen our recommendations."

Reviewer 3 Report

These recommendations have obvious limitations that were adequately  addressed by authors on discussion section. However, even taking these limitations into account, I think these recommendations are a reliable and needed guide on the use of thrombopoietin receptor agonists in patients with CLD undergoing invasive procedures.

Author Response

We thank Reviewer 3 for their comments. No changes were requested.